# Warm-Up Improves Balance Control Differently in the Dominant and Non-Dominant Leg in Young Sportsmen According to Their Experience in Asymmetric or Symmetric Sports

**DOI:** 10.3390/ijerph19084562

**Published:** 2022-04-10

**Authors:** Alfredo Brighenti, Frédéric Noé, Federico Stella, Federico Schena, Laurent Mourot

**Affiliations:** 1EA3920 Prognostic Factors and Regulatory Factors of Cardiac and Vascular Pathologies, Exercise Performance Health Innovation (EPHI) Platform, University of Bourgogne Franche-Comté, 25000 Besançon, France; alfredo.brighenti@gmail.com; 2CeRiSM, Sport Mountain and Health Research Centre, University of Verona, 38068 Rovereto, Italy; federico.stella91@gmail.com (F.S.); federico.schena@univr.it (F.S.); 3Department of Neurosciences, Biomedicine and Movement Sciences, University of Verona, 37131 Verona, Italy; 4Laboratoire Mouvement, Equilibre, Performance et Santé (EA 4445), Université de Pau et des Pays de l’Adour/E2S UPPA, 65000 Tarbes, France; frederic.noe@univ-pau.fr; 5Division for Physical Education, National Research Tomsk Polytechnic University, 634040 Tomsk, Russia

**Keywords:** acute exercise, posture, cerebral dominance, motor control laterality, postural asymmetry factor

## Abstract

The aim of this study was to investigate the acute effects of a warm-up on balance control and inter-limb balance asymmetries by analyzing the influence of the nature of the sport practiced by participants. Twelve sportspeople were recruited. They had to stand on a force plate for 30 s in a one-leg stance on their dominant (used to perform skilled movements) and non-dominant leg (used to support the body) before and 2, 5, 10, 15 and 20 min after a 10 min warm-up exercise performed at moderate intensity on a cycle ergometer. The center of foot pressure displacements was recorded. Statistical analysis was performed by considering one group of all participants and with two subgroups according to the symmetrical or asymmetrical nature of the sport they practiced. The warm-up exercise improved acute balance control only on the dominant leg after a 20 min rest without significantly reducing inter-limb balance asymmetries. This effect was more characteristic of participants with experience in asymmetric sports. These results confirm previous findings of the greater sensitivity of the dominant leg to the physiological state and reveal that between-leg differences in balance control appear mainly in subjects with experience in asymmetric sports in a specific physiological condition (post-warm-up state).

## 1. Introduction

Balance control is fundamental in activities of daily living. It is, however, a complex motor task with sensory, central and motor components that enable the integration of somatosensory, vestibular and visual cues to adequately adjust muscle activity and joint position in order to maintain the center of mass inside the base of the support’s area [1]. During sports practice in which phases of one-leg support are frequent in order to give a kick, control a ball or jump (e.g., soccer or handball), the ability to efficiently control balance from a unilateral stance is of critical importance both to optimize sports-specific performance and to limit the risk of injury [2]. Several studies have reported differences between the left and right legs when subjects stand in a unilateral stance, which reflects a bilateral asymmetry in the control of the balance between the dominant and non-dominant leg [3,4,5]. The non-dominant leg (ND-leg) is preferentially used to maintain balance from a single leg stance, while the dominant leg (D-leg) is usually used to perform skilled lower limb movements (e.g., kicking a ball or tracing shapes) [6,7,8]. Inter-limb asymmetries in balance control are characteristics of lateralization in the control of human movements. They may originate from hemisphere specialization for the utilization of somatosensory cues [9,10], which is characterized by the superiority of the non-dominant hemisphere (the right hemisphere for right-handed individuals) in processing somatosensory information, thus resulting in better proprioception in the ND-leg (i.e., the left limb for right-handed individuals) than in the D-leg [11,12]. Experience in asymmetric sports (with asymmetric technical movements/stance) [6,8] and/or specific psychophysiological states (e.g., fatigue) can accentuate this difference between the two legs [13,14,15,16]. The D-leg would be indeed more sensitive than the ND-leg to fatiguing exercises (i.e., exercises that induce decrements in the production of maximal force or power by the muscles) [14,16], especially in individuals who practice asymmetric sports [14,15], thus increasing inter-limb balance asymmetries, which can have a negative impact on sports performance and are associated with a high incidence of lower-limb injury [8,14,17,18].

Training interventions can be implemented (i.e., specific balance training, resistance training, injury prevention neuromuscular warm-up programs) to chronically improve balance control and reduce neuromuscular inter-limb asymmetries [19,20,21,22]. However, warm-up routines that incorporate moderate non-fatiguing exercises to optimize motor performance and prevent sport-related injuries can also acutely improve balance control [23,24,25,26].

Warm-up enhances the efficiency of the somatosensory system thanks to an increased sensibility of cutaneous, muscular, articular and tendinous mechanoreceptors [23]. Warm-up also enhances cortical excitability [27] and positively influences the functioning of the muscular system by enhancing muscle fiber conduction time [28,29] and increasing muscle metabolism and muscle fiber performance [30]. Hence, warm-up improves both the sensory and motor components of the balance control system [23,26]. Knowing that a suboptimal psychophysiological state such as fatigue can increase inter-limb balance asymmetries [13,14,15,16], an optimized psychophysiological state due to the performance of a warm-up routine can offer a potential benefit in reducing inter-limb balance asymmetries, especially in individuals with marked asymmetries such as those involved in asymmetrical sports. Even though the chronic effects of a long-term warm-up program have been widely investigated (e.g., [20,21]), no study, to the best of our knowledge, has specifically investigated the acute effects of a single warm-up session on inter-limb balance asymmetries.

Hence, this exploratory study was undertaken to investigate the acute effects of a warm-up routine on inter-limb balance asymmetries in young healthy sportspeople. It was hypothesized that warm-up would reduce inter-limb balance asymmetries, with more pronounced effects in individuals with experience in asymmetric sports.

## 2. Material and Methods

### 2.1. Participants

Twelve healthy sports science students (8 males, 4 females) voluntarily participated in the study (age: 22 ± 3.5 years, 19 to 31; weight 65.7 ± 9.6 kg; height: 173.4 ± 11.1 cm; mean ± SD). All participants had at least two years of experience with a minimum of 2 sessions per week in different sports disciplines of a symmetrical or an asymmetrical nature (Table 1). The exclusion criteria included any balance impairment or reported injuries in the last two years. Participants were asked to avoid intense activity and not to drink coffee and alcohol 24 h before the experimental sessions. All the participants were fully informed of the procedures and gave their written informed consent prior to participation in the study, which was approved by the Institutional Review Board in accordance with the Declaration of Helsinki.

### 2.2. Procedure

Participants went to the laboratory facilities twice, at intervals of at least 48 h (and not more than 96 h). They first underwent a familiarization session, during which they were informed of all the procedures, and they practiced the balance assessment to avoid any learning effect during the experimental session [31]. During this familiarization session, participants were also asked to pedal on a cycle ergometer for 10 min (Monark Ergomedic 839 E; Monark, Varberg, Sweden) at a fixed 80 rpm pedaling frequency and at a moderate self-regulated intensity of 20 on Borg’s scale (CR-100). The average power developed during this familiarization exercise (Pw) was collected by a power-meter (SRM^®^ Schoberer Rad Messtechnik, Julich, Germany) to be used as a target intensity in the warm-up exercise implemented in the experimental session.

The experimental session consisted of a baseline balance control assessment (pre) followed by a warm-up exercise, which consisted of pedaling for 10 min on assessed the cycle ergometer at Pw at an 80 rpm pedaling frequency. Balance control was then assessed 2 min (p2), 5 min (p5), 10 min (p10), 15 min (p15) and 20 min (p20) after completion of the warm-up. We waited 2 min before performing the first post-exercise assessment so that the subjects could recover in order to limit the influence of the increased activity of the cardiac and respiratory musculatures due to the completion of the warm-up exercise, which deforms the body and is likely to increase postural sway. P2, p5, p10 and p15 assessment times are similar to those used by Paillard et al. [26] in their study about the effects of warm-up on balance control [26]. An additional evaluation was performed 20 min after the warm-up exercise to more accurately assess the duration of the effects of the warm-up. 

Balance control was assessed with a one-leg quiet stance paradigm. Participants were asked to stand barefoot and as motionless as possible for 30 sec on a force platform (Type 9260AA, Kistler Instruments, Winterthur, Switzerland), which sampled the center of foot pressure (COP) displacements at 40 Hz. Participants had to keep their hands on the hips and the unsupported leg raised with a 90° joint flexion at the knee joint. They performed one trial on each leg (D-leg and ND-leg) in a randomized order and with a short rest period of about 10 s between trials. They also had to perform two familiarization trials on the D-leg and the ND-leg before data acquisition to avoid any learning effect and ensure a reliable measure with only one trial during the following assessments [31]. The D-leg was determined as the leg which was used by participants to kick a ball [7].

### 2.3. Analysis of Data

The following parameters were calculated: COP length in the mediolateral (COPml) and anteroposterior directions (COPap), total COP length (COPtot) and COP ellipse area (COParea). Within the framework of a quiet stance paradigm, the lower these parameters, the lower amount of postural sway and the more efficient the balance control [32]. An asymmetry index (ASI) between both legs was calculated for each COP parameter using the following formula [16]:ASI = ABS [(D-Leg − ND-Leg)/(D-Leg + ND-Leg)] × 100

An index value of zero indicates that there is no difference between the two legs, whereas a higher percentage indicates higher asymmetry.

### 2.4. Statistical Analysis

Normality was tested using the Shapiro–Wilk test. As the dependent variables did not meet the assumption of normal distribution, nonparametric tests were used. Data were analyzed in two steps. The effects of warm-up were first analyzed by considering a single group including all participants (whole group analysis) while applying Friedman’s tests on D-leg and ND-leg COP parameters (i.e., COPml, COPap, COPtot, COParea) and ASI indices (ASI_COPml, ASI_COPap, ASI_COPtot, ASI_COParea) at different assessment times (i.e., pre, p2, p5, p10, p15 and p20). When significant effects occurred, pairwise comparisons were performed with the Nemenyi test [26,33]. Differences between the D-leg and ND-leg were also tested at each assessment time with a Wilcoxon signed-rank test.

The influence of the nature of the sport practiced by participants was also analyzed by splitting the whole group into two subgroups based on the asymmetrical (ASYM) or symmetrical (SYM) nature of their sport, according to the categorization of motor tasks in sports proposed by Maloney [34]. The SYM subgroup (*n* = 7) included specialists in triathlon, running and climbing, whereas the ASYM subgroup (*n* = 5) included specialists in team and combat sports (handball, soccer, rugby, judo). Friedman’s tests, followed by the Nemenyi test, were then applied independently in each subgroup to test for a warm-up effect [26]. Potential differences between the ASYM and SYM subgroups were also tested at each assessment time with the Mann–Whitney U test. Statistical analyses were performed with R statistical software. The significance level was set at *p* < 0.05.

## 3. Results

### 3.1. Whole Group

Figure 1 illustrates the evolution of COP variables of the D-Leg and ND-leg from pre (baseline assessment, i.e., before performing the warm-up exercise) to p2, p5, p10, p15, p20 (i.e., 2 min, 5 min, 10 min, 15 min and 20 min after the completion of the warm-up). There was a significant warm-up effect on the D-leg for all the COP parameters except for COParea (COPtot: χ^2^ = 16.40; *p* = 0.006, COPap: χ^2^ = 15.52, *p* = 0.008; COPml: χ^2^ = 12.97, *p* = 0.024). Multiple comparisons revealed that these COP parameters decreased between pre and p20 (COPtot, *p* = 0.005; COPap, *p* = 0.026; COPml, *p* < 0.008). Significant differences were observed between the D and the ND-leg 20 min after the warm-up in all COP parameters (COPtot: *V* = 1, *p* < 0.001; COPap: *V* = 3, *p* = 0.002; COPml: *V* = 7, *p* = 0.001; COParea: *V* = 10, *p* = 0.021). These results mean that balance control was improved only on the D-Leg 20 min after the warm-up, thus showing that it was necessary to introduce a recovery phase to improve balance control after a warm-up exercise. There was no significant effect of the warm-up exercise on the ND-leg, which suggests that the D-leg would be more sensitive than the ND-leg to the warm-up exercise. There were no significant differences attributed to the warm-up exercise on asymmetry indexes (Figure 2). This result shows that warm-up did not acutely reduce inter-limb balance asymmetries.

### 3.2. Subgroups

Table 2 and Table 3 provide a summary of descriptive statistics and results of COP variables of the SYM and ASYM subgroups on the D-Leg and ND-leg, respectively. Results of ASY indexes in the SYM and ASYM subgroups are illustrated in Table 4. The results of the Friedman test were not significant for either the SYM or the ASYM subgroup, but strong tendencies could be observed on the D-leg in the ASYM subgroup (COPtot: χ^2^ = 10.49; *p* = 0.063; COPml; χ^2^ = 10.60; *p* = 0.059, COPap: χ^2^ = 10.26, *p* = 0.068). Multiple comparisons also revealed a significant decrease in COPml between pre and p20 (*p* < 0.047) in this subgroup (COPtot and COPap only tended to be reduced from pre to p20; *p* = 0.074). A tendency was also observed with the Freidman test on the D-leg in the SYM subgroup (COPtot: χ^2^ = 10.95; *p* = 0.052), but pairwise comparisons did not show any significant effect or tendency. These results show that warm-up-related balance improvements observed at p20 on the D-leg were more characteristic of participants from the ASYM subgroup. These effects of warm-up on the D-leg at p20 in the ASYM subgroup could explain the significant differences observed between both group at p20 in COPml (*W* = 4, *p* = 0.030) and ASI_COPml (*W* = 33, *p* = 0.010).

## 4. Discussion

The aim of this exploratory study was to investigate the acute effects of a warm-up routine on inter-limb balance asymmetries while addressing the issue of the influence of the nature of the sport practiced by participants. We hypothesized that warm-up would reduce inter-limb balance asymmetries, with more pronounced effects in individuals with experience in asymmetric sports. These hypotheses were not confirmed since the warm-up routine did not significantly reduce inter-limb balance asymmetries. Nevertheless, results showed that balance control improved after the warm-up exercise only on the D-Leg. This effect was more characteristic of participants with experience in asymmetric sports.

Our results are in line with previous studies about the effects of warm-up, which showed that balance control was improved when a recovery [26] or a stretching phase [23,24,25] was introduced after dynamic cardiovascular work. If there is no recovery or if the recovery after the dynamic phase of the warm-up routine is too short, a warm-up routine does not positively influence balance control [23,24,26,35]. The dynamic phase of a warm-up routine increases movements of internal masses in motion within the body (e.g., cardiac respiratory muscles and blood movements), which can act as an internal mechanical disturbance and amplify postural sway [26]. As postulated by Paillard et al. [26], these short-term disturbing effects should be counteracted by the temperature-related beneficial effects of the warm-up exercise, thus explaining why balance control is usually not disturbed immediately after the completion of the dynamic phase of the warm-up routine. After a few minutes of recovery or stretching following the end of the dynamic phase of a warm-up routine, the balance between the disturbing and beneficial effects of the warm-up tends to change, which leads to improved balance control [26]. Our results are in line with these previous studies. Nevertheless, we did not observe a kinetic of balance control improvements similar to that reported by Paillard et al. [26], who showed that balance control was improved 10 or 15 min after warming up on the ND-leg and D-leg, respectively (in the present study, balance control was improved 20 min after the warm-up). The more precocious beneficial effects of warm-up reported by Paillard et al. [26] can be explained by the incremental characteristic of their warm-up routine (5 min of pedaling at a heart rate of 130 b∙min^−1^, 5 min of pedaling at 150 b∙min^−1^ and 2 of pedaling min at 170 b∙min^−1^). The last 2 min of high-intensity effort of their warm-up routine might have engendered a phenomenon of post-activation potentiation (PAP), an effect which, typically expressed in a 10/15 min after the effort time window [36], may have positively influenced the motor component of the balance control system on both the D and ND-legs.

However, the results of our study are original in that only the D-leg was positively impacted by the warm-up exercise. Daneshjoo et al. [20] previously reported concordant findings while showing that the implementation of a 2-month warm-up program improved proprioception only on the D-leg in young professional soccer players. Studies about the effect of fatiguing exercises on one-leg balance control also showed that the effects of fatigue were stronger in the D-leg than in the ND-leg [13,37,38,39]. Promsri et al. [5] postulated a different sensorimotor control due to leg dominance by showing that postural movements were more tightly controlled with controller interventions that took place more regularly on the ND-leg than on the D-leg. The ability to adapt to the environmental changes is increased when controller interventions take place less frequently and less regularly [40] as in the D-leg, thus explaining why the D-leg would be more sensitive than the ND-leg to the physiological state of the subjects. When the whole sample was split into SYM and ASYM subgroups according to participants’ experience in symmetric or asymmetric sports, the results showed that post-warm-up balance improvements in the D-leg were more characteristic of participants from the ASYM subgroup. Indeed, there was a significant decrease in COPml between pre and p20 (associated with strong tendencies to a decrease in COPtot and COPap) only in the ASYM subgroup, which induced significant differences in COPml and ASI_COPml between both subgroups at p20. Paillard and Noé [8] assumed that the differences in balance control between the D-leg and the ND-leg might emerge from the interaction of multiple factors such as participants’ physiological state, sports specialty, expertise level in sports and methods of assessing the balance control. Our results seem to confirm this hypothesis by showing that between-leg differences are subtle and appear mainly in subjects with experience in asymmetric sports in a specific physiological condition, namely a post-warm-up state.

Contrary to our hypothesis, the warm-up exercise did not significantly reduce inter-limb balance asymmetries. This result reflects the inability of a warm-up routine to acutely reduce inter-limb balance asymmetries and differs from the recent studies of Madruga-Parera et al. [22] and Pardos-Mainer et al. [21], which showed that long-term interventions (resistance training and injury prevention warm-up programs) could reduce neuromuscular inter-limb asymmetries. The inability of the warm-up exercise to reduce inter-limb balance asymmetries could stem from low levels of balance asymmetries at baseline (pre), with no significant difference between the D-leg and the ND-leg both for the whole sample and in each of the two subgroups (SYM and ASYM). If participants had higher levels of asymmetry at baseline, stronger effects of the warm-up exercises on balance asymmetries would have been expected. Although we expected individuals who practiced asymmetric sports to have more pronounced asymmetries at baseline than participants involved in symmetric sports activities, we also showed that there were no significant differences between the D-leg and the ND-leg for either the SYM or the ASYM subgroup at baseline. The repetition of unilateral actions in asymmetric sports such as team sports can accentuate inter-limb balance asymmetries [3,6,41]. Nevertheless, inter-limb balance asymmetries would be heightened only in professional/expert sportsmen [8], which could explain why participants from the present study involved in asymmetrical sports (which were all sports-science students practicing several activities as part of their university education) did not present high inter-limb balance asymmetries. Future studies should be conducted with a larger sample size by including expert subjects having a more exclusive sporting practice in activities associated with an asymmetrical or symmetrical motricity in order to analyze the influence of motor experience on the ability to benefit from a warm-up routine to reduce inter-limb balance asymmetries.

The main limitation of this study was the small sample size. Given the size limitations and exploratory nature of this study, caution in the generalization of these results has to be observed, and future studies deserve to be conducted with a larger sample size to confirm these results. It should also be noted that very few studies investigated the influence of gender on inter-limb balance asymmetries with inconsistent results (e.g., [42,43]). Even though both male and female participants were included in the present study, we could not perform any between-gender comparison because of the small sample size. Hence, further experiments are also needed to determine whether gender can influence the effects of warm-up on inter-limb balance asymmetries and, more generally, to clarify the influence of gender on asymmetries in balance control. Further studies are also warranted to analyze the influence of anthropometric factors such as body weight, height, muscle mass and morphological asymmetry. Similarly, the effects of age and the number of years of sports practice should also be investigated.

## 5. Conclusions

A warm-up routine that involved the performance of a moderate self-regulated intensity exercise did not acutely reduce inter-limb balance asymmetries in young, healthy sportspeople, regardless of the nature of the sport they practice (symmetric or asymmetric). Nevertheless, a warm-up could improve one-leg balance control on the D-leg when a 20 min recovery period is introduced after the end of the warm-up, thus illustrating the need for a short period of recovery after the warm-up routine to optimize balance control. This beneficial effect of warming up on the D-leg was more likely to be observed in individuals who practice asymmetric sports. These results confirm previous findings of the D-leg’s greater sensitivity to the physiological state of the subjects and also reveal that the nature of the sport practiced by the participants is likely to modulate this greater sensitivity of the D-leg.

## Figures and Tables

**Figure 1 ijerph-19-04562-f001:**
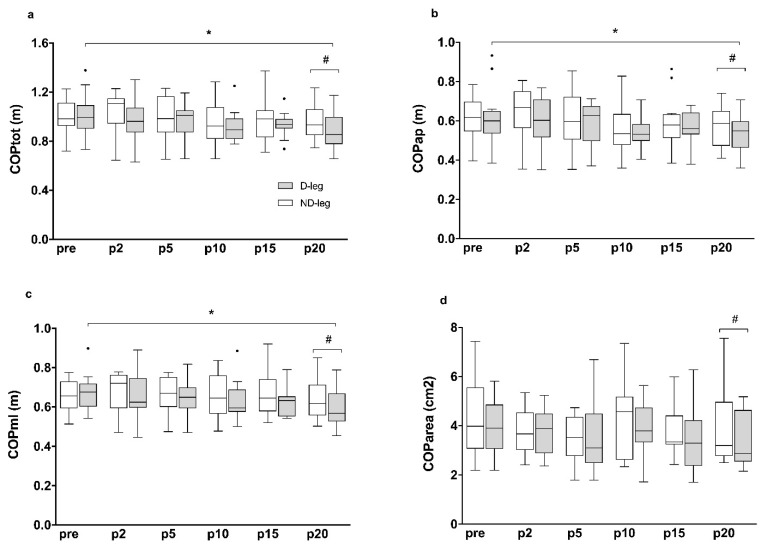
Boxplot representation of COP parameters in the D-leg and the ND-leg before (pre) and after (p2, p5, p10, p15 and p20) performing the warm-up routine. COPtot: total COP length (**a**); COPap: COP length in the anteroposterior direction (**b**); COPml: COP length in the mediolateral direction (**c**); COParea: COP ellipse area (**d**); Assessments were performed before (pre), 2 min (p2), 5 min (p5), 10 min (p10), 15 min (p15) and 20 min (p20) after the completion of the warm-up. Small circles indicate data points beyond the whiskers. *: significant difference between two assessment times; #: significant difference between the D-leg and the ND-leg (*p* < 0.05).

**Figure 2 ijerph-19-04562-f002:**
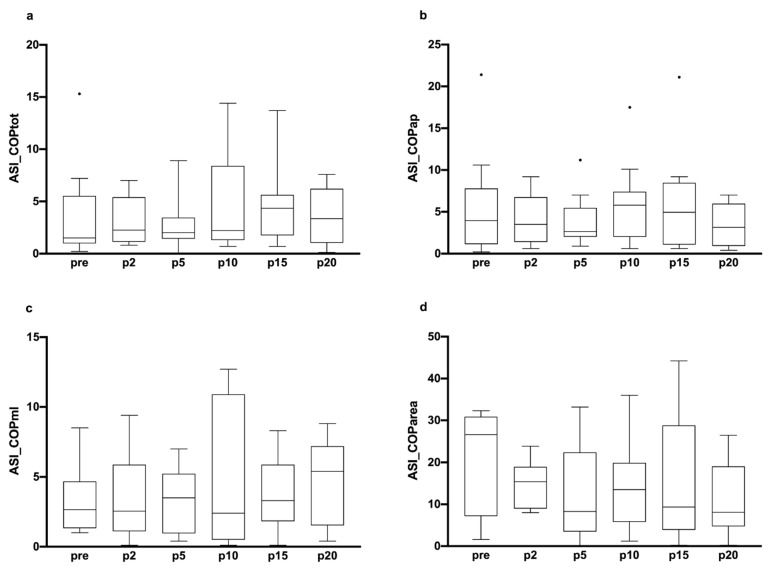
Boxplot representation of asymmetry indexes before (pre) and after (p2, p5, p10, p15 and p20) performing the warm-up routine. ASI: asymmetry indexes between both legs calculated for the following COP parameter: COPtot (total COP length; (**a**)), COPap (COP length in the anteroposterior direction; (**b**)), COPml (COP length in the mediolateral direction; (**c**)) and COParea (COP ellipse area; (**d**)). Assessments were performed before (pre), 2 min (p2), 5 min (p5), 10 min (p10), 15 min (p15) and 20 min (p20) after the completion of the warm-up. Small circles indicate data points beyond the whiskers.

**Table 1 ijerph-19-04562-t001:** Participants’ characteristics and experience in sports.

Subject	Sex	Age	Height	Weight	Sports Specialty	Experience	Subgroup
1	F	26	177	60	Triathlon	3 years	SYM
2	M	23	175	69	Handball	8 years	ASYM
3	M	22	191	85	Triathlon	2 years	SYM
4	F	19	155	49	Rock climbing	8 years	SYM
5	M	22	172	64	Soccer	15 years	ASYM
6	F	31	165	59	Running	6 years	SYM
7	M	20	192	73	Rock climbing	6 years	SYM
8	M	20	172	63	Judo	14 years	ASYM
9	M	22	173	72	Karate	10 years	ASYM
10	M	20	180	70	Rugby	7 years	ASYM
11	F	20	159	54.5	Rock climbing	6 years	SYM
12	M	19	170	70	Running	7 years	SYM

F: female; M: male; SYM: subgroup of participants practicing a sport of a symmetric nature; ASYM: subgroup of participants practicing a sport of an asymmetric nature.

**Table 2 ijerph-19-04562-t002:** COP variables from the D-leg in the SYM and ASYM subgroups.

		Pre	p2	p5	p10	p15	p20
SYM	COPtot	982.30 (133.30)	953.97 (160.38)	1049.76 (144.94)	955.09 (141.48)	939.75 (61.40)	938.18 (227.52)
	COPap	592.57 (92.96)	579.44 (78.02)	631.03 (103.20)	556.58 (70.49)	555.44 (59.42)	578.87 (142.16)
	COPml	674.87 (76.58)	673.13 (154.03)	670.70 (99.67)	641.70 (119.00)	649.54 (48.80)	617.38 (150.73) ^†^
	COParea	409.62 (117.88)	401.01 (42.16)	383.77 (133.48)	384.65 (63.50)	305.87 (183.48)	276.22 (196.91)
ASYM	COPtot	1005.77 (97.98)	1025.95 (92.08)	918.48 (148.56)	842.15 (78.28)	934.18 (32.67)	783.06 (89.11)
	COPap	600.85 (6.23)	624.45 (162.61)	574.93 (132.64)	526.31 (39.89)	613.97 (91.78)	542.00 (97.81)
	COPml	682.44 (92.79)	623.52 (86.63)	610.28 (53.80)	575.54 (11.46)	556.11 (90.10)	523.46 (73.37) *^pre^
	COParea	388.35 (87.62)	369.83 (160.10)	242.91 (40.31)	343.70 (157.82)	352.81 (113.21)	299.83 (138.86)

Data are expressed as median (interquartile range). COPtot: total COP length; COPap: COP length in the anteroposterior direction; COPml: COP length in the mediolateral direction; COParea: COP ellipse area; SYM: subgroup of participants practicing a sport of a symmetric nature; ASYM: subgroup of participants practicing a sport of an asymmetric nature; Assessments were performed before (pre), 2 min (p2), 5 min (p5), 10 min (p10), 15 min (p15) and 20 min (p20) after the completion of the warm-up. * indicates significant differences related to the warming-up exercise; acronym in superscript indicates the time in which there were significant differences; ^†^ indicates significant between-group differences at a specific time (*p* < 0.05).

**Table 3 ijerph-19-04562-t003:** COP variables from the ND-leg in the SYM and ASYM subgroups.

		Pre	p2	p5	p10	p15	p20
SYM	COPtot	1080.07 (169.10)	1115.19 (180.40)	1016.68 (254.74)	1043.62 (305.71)	1032.11 (188.97)	969.80 (195.58)
	COPap	669.17 (119.12)	677.56 (122.39)	622.45 (157.82)	630.58 (208.82)	621.60 (176.88)	604.71 (135.20)
	COPml	682.63 (129.61)	752.29 (92.87)	682.05 (156.29)	698.51 (156.47)	691.88 (98.72)	649.44 (116.91)
	COParea	470.47 (262.56)	334.81 (77.40)	328.08 (129.09)	436.86 (90.68)	330.37 (39.98)	296.17 (209.68)
ASYM	COPtot	962.55 (54.06)	975.62 (179.07)	951.15 (199.87)	875.45 (103.47)	881.56 (128.87)	881.99 (48.01)
	COPap	598.58 (43.62)	612.17 (130.26)	559.82 (126.23)	532.38 (56.24)	551.14 (49.91)	548.41 (115.79)
	COPml	626.69 (95.18)	685.43 (126.39)	655.94 (145.60)	581.71 (81.24)	576.44 (109.18)	596.69 (59.87)
	COParea	371.02 (94.15)	386.31 (93.95)	368.99 (69.48)	457.30 (261.18)	440.39 (117.62)	330.07 (171.60)

Data are expressed as median (interquartile range). SYM: subgroup of participants practicing a sport of a symmetric nature; ASYM: subgroup of participants practicing a sport of an asymmetric nature; COPtot: total COP length; COPap: COP length in the anteroposterior direction; COPml: COP length in the mediolateral direction; COParea: COP ellipse area. Assessments were performed before (pre), 2 min (p2), 5 min (p5), 10 min (p10), 15 min (p15) and 20 min (p20) after the completion of the warm-up. There were no significant differences related to the warming-up exercise and no significant between-group differences at any time.

**Table 4 ijerph-19-04562-t004:** ASI indexes in the SYM and ASYM subgroups.

		Pre	p2	p5	p10	p15	p20
SYM	ASI_COPtot	1.79 (3.85)	2.19 (1.10)	2.26 (1.73)	1.49 (1.19)	4.68 (5.46)	1.66 (3.29)
	ASI_COPap	4.47 (5.44)	4.09 (4.27)	3.79 (3.45)	4.88 (3.20)	4.25 (4.6)	2.24 (3.32)
	ASI_COPml	1.52 (3.18)	3.20 (3.83)	3.39 (3.66)	2.29 (0.55)	3.87 (2.93)	2.77 (3.81) ^†^
	ASI_COParea	29.44 (19.43)	17.68 (2.24)	7.47 (7.50)	13.26 (14.65)	15.93 (23.66)	9.16 (15.93)
ASYM	ASI_COPtot	1.20 (0.34)	2.33 (4.15)	1.75 (2.60)	6.96 (3.75)	3.97 (1.99)	5.94 (1.60)
	ASI_COPap	3.42 (3.24)	2.93 (3.71)	2.62 (0.32)	5.78 (5.37)	5.94 (7.24)	4.06 (5.07)
	ASI_COPml	2.74 (0.92)	2.39 (1.55)	3.64 (2.90)	9.35 (10.86)	2.58 (0.55)	7.18 (0.62)
	ASI_COParea	23.87 (22.11)	10.24 (6.26)	11.60 (21.07)	16.15 (15.90)	6.52 (4.21)	4.80 (3.93)

Data are expressed as median (interquartile range). SYM: subgroup of participants practicing a sport of a symmetric nature; ASYM: subgroup of participants practicing a sport of an asymmetric nature; ASI: asymmetry indexes between both legs calculated for the following COP parameter: COPtot (total COP length), COPap (COP length in the anteroposterior direction), COPml (COP length in the mediolateral direction) and COParea (COP ellipse area). Assessments were performed before (pre), 2 min (p2), 5 min (p5), 10 min (p10), 15 min (p15) and 20 min (p20) after the completion of the warm-up. ^†^ indicates significant between-group differences at a specific time (*p* < 0.05).

## Data Availability

Original data will be made available to any qualified researcher upon request.

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
