# Peer review of "Warm-Up Improves Balance Control Differently in the Dominant and Non-Dominant Leg in Young Sportsmen According to Their Experience in Asymmetric or Symmetric Sports"

_ijerph, 2022, doi:10.3390/ijerph19084562_

Round 1

Reviewer 1 Report

This study used 12 particpants to investigate the effects of warm-up on balance control and inter-limb balance asymmetries by analyzing the influence of the nature of the sport. Results partly support their claim but I have a few questions:

1) the biggest issue of this study was insufficient sample. They only recruited 12 participants and this number was not justified by power calculation. Small sample would have great influence on the results and conclusions. Also, because of the small sample, the distribution of the data was not normal so that they have to use non-parametric statistics. 

2) the second issue was associated with the first one, they only recruited the sports specialists, and they should have one group of normal participants so that they can conclude on the influence of sport training.

3) some of the writing style needs to check. For example, "investi-gated" in Abstract should be spelled as "investigated".

Reviewer 2 Report

The manuscript entitled “Warm-up improves balance control differently in the dominant and non[1]dominant leg in young sportsmen according to their experience in asymmetric or symmetric sports “ was reviewed as stated further. This study investigated the effects of warm-up on balance control and inter-limb balance asymmetries by analyzing the influence of the nature of the sport practiced by young healthy participants. This manuscript require some minor changes to be suitable for publication in the Journal. My comments on this manuscript are formulated as follows:

Abstract

Line 2: change “investi-gated” to “investigated”

Line 3: change “in-ter-limb” to “inter-limb”

Line 4: Change “Twelve participants were recruited. They were grouped depending on their experi-ence in symmetrical or asymmetrical sports“ to “Twelve participants were recruited, and grouped depending on their experience in symmetrical or asymmetrical sports“.

Line 8: change „im-proved“ to “improved”

Line 10: change „ experi-ence“ to “experience”

Line 12: change „ ap-pear “ to “appear”

  1. Introduction

This part has satisfactorily been written. In my opinion the need to perform this study was sufficiently explained.

  1. Materials and methods

In general, the methods section has been written with all required details. However, the below changes must be applied.

Section 2.1:  change „prior the participation on the study“ to “prior to the participation in the study“

Section 2.2 (2nd paragraph): change “consisted in a baseline balance“ to “consisted of a baseline balance“

  1. Results

The number of male and female participants are not equal but I wonder if there is any comparison between genders on COP and other measurements?

  1. Discussion

How was the experience of the participants evaluated bearing in mind that they do different sports and obviously different muscle types are trained.

Reviewer 3 Report

General comments

The topic of the paper is interesting and fits the scope of the journal. The text is relatively well written and composed. However, I have severe reservations about the methodology and results presented. For example, the sample size is very small.

Specific comments

Please add a table with anthropometric characteristics of subjects.

The warm up included only 10 minutes on the cycle ergometer or and stretching?

Please explain why did you choose 2 minutes, 5 minutes, 10 minutes, 15 minutes and 20 minutes after the warm up?

Please refer which was the time rest between the trials of dominant leg and not dominant leg?

How many balance trials did the athletes with each leg?

Reviewer 4 Report

Dear Authors,
Thank you for the opportunity to review this paper. I my opinion, the manuscript describes a significant problem but the study group is too small to draw meaningful conclusions. 

The comments are listed below:
Major comments : 
1.Whether this manuscript is in line with the journal's profile - I leave the decisions to the Editor.
2. The theoretical sections are well written, the methods have been selected correctly, but the study group is small, with a large age range, additionally 
both genders have been tested and there is no control group. 
3. The body composition parameters of the subjects were not given. The level of morphological asymmetry (e.g., length of the lower leg, length of the dominant leg versus the non-dominant leg; appropriate circumferences) may be important for inferring functional / dynamic asymmetry.

Minor comments :
1. Keywords should be different than the words in the title.
2. The number of decimal places (e.g. at p values) should be standardized.

Thank you.

Reviewer 5 Report

The paper must be proofread by an English native speaker. Please give more details about study participants (n=12): Age? Sex? BMD? Sport? Weight? Height? Muscle mass? Number of years of sports practice? etc. Obtained findings per study participant?

Abstracts:

- Spelling: “investi-gated”; “experi-ence”, “im-proved”, “experi-ence”, “ap-pear”. Please check.

- “dominant leg (D-leg)”; please define all concepts when first introduced in the text

- “has never been investi-gated”. Are you sure? Have you carried out a review? Please rewrite this sentence.

- Please see these papers: https://scholar.google.com/scholar?hl=pt-PT&as_sdt=0%2C5&q=dominat+leg+warm+up&btnG=

- “symmetrical or asymmetrical sports”; please define all concepts when first introduced in the text

-“warm-up exercise”; please give more details.

- Please rewrite this sentence: “Statistics were performed on the whole sample and the two groups of individuals”. “on the whole sample”?

Keywords: please use some MeSH terms. Please see: https://www.ncbi.nlm.nih.gov/mesh/

  1. Introduction

- Please define all concepts when first introduced in the text. For instance, “Experience in asymmetric sports [6,8]”. “asymmetric sports”, “fatiguing exercises [14,16]”, “warm-up routines”, etc. All concepts should be explained when first introduced in the text.

- Introduction, first paragraph: “During sports practice in which phases of one-leg support are frequent”. Please give examples of sports practice.

- The first paragraph is too long.

- “Nevertheless, to our knowledge, no study has specifically investigated the acute effects of warm-up on inter-limb balance asymmetries.” Have you carried out a review? Are you sure?

- Please cite more similar or related works. Please see these papers: https://scholar.google.com/scholar?hl=pt-PT&as_sdt=0%2C5&q=dominat+leg+warm+up&btnG=

- International guidelines/recommendations? Please cite international guidelines on the present topic.

- “It was hypothesized”; Is this study representative?

  1. Material and methods

  • Please give more details: Who? Where? When? Followed equator guidelines?
  • Please see https://www.equator-network.org/.

2.1. Participants

- “All participants were specialists in different sports”? “specialists”? Please rewrite.

2.4. Statistical analysis

- Please cite studies applying similar statistical methodologies

  1. Results

- Please create subheadings in results. Results are too compact.

- Please create a section about participants data. Age? Sex? BMD? Sport? Weight? Height? Muscle mass? Number of years of sports practice? etc.

- Please briefly explain all Figures and Tables in the section of results; please see the comments about Tables and Figures. Please give more details in results.

- Please check the presentation of statistical findings in APA guidelines.

Discussion

- Interpretation/explanation of study findings should be placed in results.

- Explanations about study hypothesis should be placed at the end of Discussion.

- Please cite more similar or related studies in discussion; you may use the new cited references in introduction.

- Please discuss study findings per participants data, i.e., “Age? Sex? BMD? Sport? Weight? Height? Muscle mass? Number of years of sports practice? etc.”

- For instance, [23,24,25] or [23-25]? Please check the formats of all references. Please see instructions for authors.

- Please create a section about practical implications, future research, and study limitations at the end of Discussion.

Conclusion

Conclusion should reply to “this exploratory study was undertaken to investigate the acute effects of a warm-up routine on inter-limb balance asymmetries in young healthy sportspeople”. Please optimize study conclusion.

References

  • Please check the format of all references in instructions for authors.

Tables and Figures

  • The meaning of all abbreviations should be placed below the Table. For instance, SYM, ASYM, pre, p2, etc. in Table 1. Please check all Tables and Figures.
  • Figure 1 and 2. Please use Figure 1a.; Figure 1b.; Figure 1c; and Figure 1d. and Figure 2a.; Figure 1/2b.; Figure 2c; and Figure 2d. and explain the interpretation of all findings in results.

Round 2

Reviewer 1 Report

none

Author Response

Please see attached doc.

Reviewer 3 Report

The topic of the paper is interesting and fits the scope of the journal. The text is relatively well written and composed. However, the study group is small, with a large age range, and there is no control group. 

Author Response

Please see attached doc.

Reviewer 4 Report

Dear Authors,

I have no more comments. Thank you for taking into account the comments previously submitted.

Thank you.

Author Response

Please see attached doc.

Reviewer 5 Report

Congratulations, the quality of the paper has been improved. Please present the full meaning of all abbreviations when they are first presented in the text (e.g., COPap; COPtot, etc.). Please check formats, for instance “3- 1- Whole group” or “3.1 Whole group”? Conclusion: please place the word “acute” in introduction.

Author Response

Please see attached doc.